# The Evaluation of DHPMs as Biotoxic Agents on Pathogen Bacterial Membranes

**DOI:** 10.3390/membranes12020238

**Published:** 2022-02-18

**Authors:** Barbara Gawdzik, Paweł Kowalczyk, Dominik Koszelewski, Anna Brodzka, Joanna Masternak, Karol Kramkowski, Aleksandra Wypych, Ryszard Ostaszewski

**Affiliations:** 1Institute of Chemistry, Jan Kochanowski University, Uniwersytecka 7, 25-406 Kielce, Poland; j.masternak@ujk.edu.pl; 2Department of Animal Nutrition, The Kielanowski Institute of Animal Physiology and Nutrition, Polish Academy of Sciences, Instytucka 3, 05-110 Jabłonna, Poland; 3Institute of Organic Chemistry, Polish Academy of Sciences, Kasprzaka 44/52, 01-224 Warsaw, Poland; dominik.koszelewski@icho.edu.pl (D.K.); anna.brodzka@icho.edu.pl (A.B.); ryszard.ostaszewski@icho.edu.pl (R.O.); 4Department of Physical Chemistry, Medical University of Bialystok, Kilińskiego 1 Str., 15-089 Białystok, Poland; kkramk@wp.pl; 5Centre for Modern Interdisciplinary Technologies, Nicolaus Copernicus University in Torun, ul. Wileńska 4, 87-100 Toruń, Poland; wypych@umk.pl

**Keywords:** 3,4-dihydropyrimidin-2(1H)-ones (DHPMs) obtained by Biginelli reaction, DNA-N-glycosylase, Fpg protein formamidopyrimidine, lipopolysaccharide (LPS)

## Abstract

Herein, we present biological studies on 3,4-dihydropyrimidin-2(1H)-ones (DHPMs) obtained via Biginelli reaction catalyzed by NH4Cl under solvent-free conditions. Until now, DHPMs have not been tested for biological activity against pathogenic *E. coli* strains. We tested 16 newly synthesized DHPMs as antimicrobial agents on model *E. coli* strains (K12 and R2–R4). Preliminary cellular studies using MIC and MBC tests and digestion of Fpg after modification of bacterial DNA suggest that these compounds may have greater potential as antibacterial agents than typically used antibiotics, such as ciprofloxacin (ci), bleomycin (b) and cloxacillin (cl). The described compounds are highly specific for pathogenic *E. coli* strains based on the model strains used and may be engaged in the future as new substitutes for commonly used antibiotics in clinical and nosocomial infections in the pandemic era.

## 1. Introduction

3,4-Dihydropyrimidin-2(1H)-ones (DHPMs) are heterocycles with a pyrimidine moiety in their structure, synthesized via multicomponent Biginelli reaction [1,2] under various reaction conditions [3,4,5]. These compounds are of great importance in microbiology and medicinal chemistry due to their specific biological and pharmacological activities [6,7]. Monastrol as a specific compound is a protagonist of the DHPM class. Research has revealed that its action as an inhibitor on human kinesin Eg5 leads to apoptosis by mitotic spindle arrest [8,9,10,11]. Reported data show other possible targets for these molecules, such as centrin [12], calcium channels [13] and topoisomerase I [14]. Pharmacological properties are reported to include anticancer [15] anti-inflammatory [16], antihypertensive [17], antibacterial [18], antifungal [19], antiviral [20], antiparasitic [21], antithyroid [22], antimuscarinic [23], antidiabetic [24] and hypolipidemic [25] activities. DHPMs play an antagonistic/inhibitory role against acetylcholinesterase [26], urease [27], calcium channel modulation [28,29] and GABA agonism [30]. The general structure of these compounds is depicted in Figure 1 [31]. However, despite many literature reports, the pharmacological and biological properties of DHPMs are still being discovered in practical clinical biochemistry and microbiological engineering; therefore, there is still a need for additional research on their cytotoxic effects on selected hospital bacterial strains causing diseases associated with blood infections, such as sepsis. There is still a need for further in vivo studies to better delineate the pharmacological potential of this class of substances. Here, we present an attempt to answer this question. Until now, such compounds have not been tested for biological activities against pathogenic *Escherichia coli* strains, so there is a need to clarify their role.

## 2. Materials and Methods

### 2.1. Microorganisms and Media

Bacterial strains were obtained as a donation from Prof. Jolanta Łukasiewicz (Polish Academy of Sciences, Wrocław, Poland).

### 2.2. Agarose-Native Gel Electrophoresis

Basic protocol for agarose-native gel electrophoresis in horizontal mode is as follows, unless otherwise indicated. Agarose was dissolved at 1% in hot 0.1 M His/0.1 M MES buffer at pH 6.1 and cast onto a flat bed with a comb in the center position. After loading the sample, electrophoresis was run at room temperature for 30 min or an indicated period under a constant voltage of 100 V. Electrophoresis was run on an ATTO-AE6500 apparatus by ATTO (Tokyo, Japan).

### 2.3. Statistical Analysis

All experimental data were presented as the means ± standard error of the mean (SEM) of a minimum of three independent experiments for BeWo cells (*n* = 3). In the case of villous explants, each experiment was performed on five independent cultures of human placenta explants (*n* = 5). Normality was checked by the Shapiro–Wilk test. One-way ANOVA was used to determine differences among more than two treatment groups, and the Tukey test was used post hoc Prism 8 Software by GraphPad (La Jolla, CA, USA). Statistical significance is indicated by different letters (*p* < 0.05): the same letters indicate no significant difference, with a < b < c < d < e < f or * *p* < 0.05, ** *p* < 0.1 and *** *p* < 0.01.

### 2.4. Experimental Chemistry

All reagents used in this work were purchased from Sigma-Aldrich (Sant Louis, MO, USA). Melting points were measured on an Electrothermal 9200 apparatus (Cole-Parmer, Staffordshire, UK) and are uncorrected. ^1^H NMR, ^13^C NMR and ^19^F NMR spectra were recorded in solutions (DMSO-*d*_6_) on a Varian 400 spectrometer (Agilent, Santa Clara, CA, USA). Elemental analysis was carried out on a vario MICRO cube elemental Analyzer Elementar, (Langenselbold, Germany). FTIR spectra were recorded with a Nicolet 380 FT-IR spectrophotometer (Thermo Fisher Scientific, Waltham, MA, USA) in the region of 4000–400 cm^−1^ using the KBr discs method. HR-ESI-MS spectra were recorded on a BrukermicrOTOF-Q II (Bruker, Billerica, MA, USA) with electrospray ionization (ESI). Analytical thin-layer chromatography (TLC) was carried out on silica gel 60 F_254_ (Merck, Darmstadt, Germany), and various developing systems were applied. Compounds were detected with 254 nm UV light (Lamp UV Consort 2 × 15 W, 254 NM, VL215-C). Column chromatography was performed on silica gel (Kiselgel 60, 230–400 mesh Merck, Darmstadt, Germany) with acetone/*n*-hexane 1:2 (*v*/*v*). 

## 3. Microorganisms and Media

Similar research is presented in [32,33,34,35,36,37,38,39,40]. The newly synthesized compounds presented in our earlier manuscripts were described as new and innovative broad-spectrum antimicrobial drugs more potent than the antibiotics commonly used in nosocomial infections. All analyzed compounds belong to the compounds referred to as peptidomimetics, with a structure and function similar to that of peptides. Therefore, the analysis of new compounds from this group is extremely important in hospital or clinical infections [1,2,3,4,5,6,7,8,9,10,11,12,13,14,15,16,17,18,19,20,21,22,23,24,25,26,27,28,29,30,31,32,33,34]. On the basis of the obtained results, including MIC and MBC tests, it was found that the analyzed compounds significantly affect the defragmentation of the membrane and the structure of the cell wall of bacteria containing LPSs of various lengths [32,33,34,35,36,37,38,39,40]. Additionally, studies were carried out related to the effects of the oxidative stress generated in the cell under the influence (modification) of the analyzed compounds on the damage and modification of bacterial DNA after digestion with the Fpg enzyme (Labjot, New England Biolabs, UK). The obtained values of oxidative damage after digestion with Fpg protein were compared with the modifications to bacterial DNA after treatment with antibiotics, such as ciprofloxacin, bleomycin and cloxacillin [32,33,34,35,36,37,38,39,40]. The presented research shows that 3,4-dihydropyrimidin-2(1H)-ones (DHPMs) could be used in the future as potential “candidates” for new drugs in relation to the analyzed antibiotics [1,2,3,4,5,6,7,8,9,10,11,12,13,14,15,16,17,18,19,20,21,22,23,24,25,26,27,28,29,30,31,32,33,34,35,36,37,38,39,40,41,42,43,44,45]. 

## 4. Experimental Section

### 4.1. General Procedure for the Synthesis of P-fluorophenyl-substituted 3,4-dihydropyrimidine-2 (1H)-one Derivatives ***4***–***6***

A mixture of 3.7 mmol of 4-fluorobenzaldehyde or benzaldehyde, 5.6 mmol of urea or thiourea, 3.7 mmol acetoacetate and 1.5 mmol NH_4_Cl was heated at 100 °C and stirred for 3 h. After the completion of the reaction, as indicated by TLC (EtOAc/*n*-hexane, 1:2, *v*/*v*), 10 mL of diethyl ether was added to the reaction mixture. The precipitate was filtered and dried in vacuo. The crude product was purified by crystallization from ethanol. The crude products, **4d** and **4e**, were purified by column chromatography on silica gel using acetone/*n*-hexane 1:2 (*v*/*v*) as an eluent. 

### 4.2. Product ***4a***: Ethyl 4-(4-fluorophenyl)-6-methyl-1,2,3,4-tetrahydro-2-oxopyrimidine-5-carboxylate

The crude product was purified by crystallization from ethanol to obtain a suitable product in the form of colorless crystals; mp. 179–180 °C [Lit mp. 181–183 °C] [46]; ^1^H NMR (400 MHz, DMSO-*d*_6_) δ 9.17 (s, 1H), 7.70 (d, *J* = 2.9 Hz, 1H), 7.24 (dd, *J* = 8.5, 5.5 Hz, 2H), 7.12 (t, *J* = 8.7 Hz, 2H), 5.13 (d, *J* = 3.3 Hz, 1H), 4.07–e3.87 (m, 2H), 2.23 (s, 3H), 1.07 (t, *J* = 7.1 Hz, 3H); ^13^C NMR (100 MHz, DMSO-*d*_6_) δ 165.7, 152.4, 148.9, 141.6, 128.7, 128.6, 115.6, 115.4, 99.6, 59.6, 53.8, 18.2, 14.5; ^19^F NMR (376 MHz, DMSO-*d*_6_) δ −115.47. ^1^H- and ^13^CNMR data were in accordance with those reported in the literature [47]; IR (cm^−1^): 3236, 2977, 1704, 1651, 1461, 1219, 1083, 816, 794, 605; ESI-TOF HR: *m*/*z* calculated for C_14_H_15_FN_2_O_3_: 557.2212 [2M+H]^+^, found 557.2233; elemental analysis calculated for C_14_H_15_FN_2_O_3_: C, 60.43%; H, 5.43%; N, 10.07%, found C, 59.72%; H, 5.44%; N, 10,06.

### 4.3. Product ***4b***: Methyl 4-(4-fluorophenyl)-6-methyl-1,2,3,4-tetrahydro-2-oxopyrimidine-5-carboxylate

The crude product was purified by crystallization from ethanol to obtain a suitable product in the form of colorless crystals; mp. 195–196 °C [Lit mp.191–192 °C] [48]; ^1^H NMR (400 MHz, DMSO-*d*_6_) δ 9.28–9.06 (m, 1H), 7.72 (d, *J* = 3.5 Hz, 1H), 7.31–7.21 (m, 2H), 7.12 (t, *J* = 8.9 Hz, 2H), 5.13 (d, *J* = 3.4 Hz, 1H), 3.51 (s, 3H), 2.24 (s, 3H); ^13^C NMR (100 MHz, DMSO-*d*_6_) δ 166.2, 152.4, 149.2, 141.4, 128.6, 128.5, 115.7, 115.5, 99.4, 53.6, 51.2, 18.2; ^19^F NMR (376 MHz, DMSO-*d*_6_) δ -115.4. ^1^H- and ^13^CNMR data were in accordance with those reported in the literature [49]; IR (cm^−1^): 3327, 2952, 1697, 1667, 1414, 1324, 1240, 1094, 825, 780, 646; ESI-TOF HR: *m*/*z* calculated for C_13_H_13_FN_2_O_3_: 551.1718 [2M+Na]^+^, found 551.1691; elemental analysis calculated for C_13_H_13_FN_2_O_3_: C, 59.09%; H, 4.96%; N, 10.60%, found C 59.12%; H, 4.59%; N, 10.63%.

### 4.4. Product ***4c***: iso-Butyl 4-(4-fluorophenyl)-6-methyl-1,2,3,4-tetrahydro-2-oxopyrimidine-5-carboxylate

The crude product was purified by crystallization from ethanol to obtain a suitable product in the form of white crystals; mp. 164–165 °C, ^1^H NMR (400 MHz, DMSO-*d*_6_) δ 9.22 (s, 1H), 7.71 (s, 1H), 7.24 (dd, *J* = 8.5, 5.7 Hz, 2H), 7.13 (dd, *J* = 9.9, 7.7 Hz, 2H), 5.14 (d, *J* = 3.4 Hz, 1H), 3.82–3.61 (m, 2H), 2.26 (s, 3H), 1.73 (dt, *J* = 13.2, 6.6 Hz, 1H), 0.72 (dd, *J* = 6.8, 3.6 Hz, 6H); ^13^C NMR (100 MHz, DMSO-*d*_6_) δ 165.7, 152.3, 149.4, 128.7, 128.6, 115.7, 115.4, 99.2, 69.7, 31.1, 27.6, 19.3, 19.2, 18.2; ^19^F NMR (376 MHz, DMSO-*d*_6_) δ-115.43; IR (cm^−1^): 3315, 2962, 1704, 1682, 1643, 1507, 1467, 1379, 1223, 1092, 841, 796, 662; ESI-TOF HR: *m*/*z* calculated for C_16_H_19_FN_2_O_3_: 635.2657 [2M+Na]^+^, found 635.2644; elemental analysis calculated for C_16_H_19_FN_2_O_3_: C, 62.73%; H, 6.25%; N, 9.14%, found C, 62.23%; H, 6.26%; N, 9.17%.

### 4.5. Product ***4d***: Ethyl 4-(4-fluorophenyl)-6-propyl-1,2,3,4-tetrahydro-2-oxopyrimidine-5-carboxylate

The crude product was purified by column chromatography with silica gel (*n*-hexane/acetone; 2:1; *v*/*v*) to afford the corresponding product as colorless crystals; mp. 152–153 °C, ^1^H NMR (400 MHz, DMSO-*d*_6_) δ 9.23–9.06 (m, 1H), 7.72–7.64 (m, 1H), 7.24 (dd, *J* = 8.6, 5.6 Hz, 2H), 7.12 (t, *J* = 8.9 Hz, 2H), 5.13 (d, *J* = 3.4 Hz, 1H), 3.97 (qd, *J* = 7.1, 1.6 Hz, 2H), 2.61 (td, *J* = 7.4, 4.3 Hz, 2H), 1.54 (q, *J* = 7.5 Hz, 2H), 1.07 (t, *J* = 7.1 Hz, 3H), 0.89 (t, *J* = 7.3 Hz, 3H); ^13^C NMR (100 MHz, DMSO-*d*_6_) δ 165.6, 152.6, 152.4, 128.7, 128.9, 115.7, 115.6, 99.3, 59.0, 53.9, 31.1, 22.8, 14.5, 14.0; ^19^F NMR (376 MHz, DMSO-*d*_6_) δ −115.2; IR (cm^−1^): 3441, 2966, 1698, 1644, 1508, 1456, 1232, 1095, 836, 773; ESI-TOF HR: *m*/*z* calculated for C_16_H_19_FN_2_O_3_: 305.1301 [M+H]^+^, found 305.1301; elemental analysis calculated for C_16_H_19_FN_2_O_3_: C, 62.73%; H, 6.25%; N, 9.14%, found 62.74%; H, 6.22%; N, 9,17%. 

### 4.6. Product ***4e***: Ethyl 4-(4-fluorophenyl)-6-phenyl-1,2,3,4-tetrahydro-2-oxopyrimidine-5-carboxylate

The crude product was purified by column chromatography with silica gel (*n*-hexane/acetone; 2:1; *v*/*v*) to afford the corresponding product as colorless crystals; mp. 168–169 °C [Lit mp. 168–170 °C] [50]; ^1^H NMR (400 MHz, DMSO-*d*_6_) δ 9.28 (d, *J* = 2.0 Hz, 1H), 7.82 (dd, *J* = 3.7, 1.9 Hz, 1H), 7.47–7.33 (m, 5H), 7.33–7.25 (m, 2H), 7.19 (t, *J* = 8.8 Hz, 2H), 5.23 (d, *J* = 3.4 Hz, 1H), 3.69 (q, *J* = 7.1 Hz, 2H), 0.70 (t, *J* = 7.1 Hz, 3H); ^13^C NMR (101 MHz, DMSO-*d*_6_) δ 165.5, 152.4, 149.5, 141.1, 135.4, 129.3, 128.8, 128.8, 128.7, 128.1, 115.8, 115.6, 100.7, 59.5, 53.9, 13.8; ^19^F NMR (376 MHz, DMSO-*d*_6_) δ -115.2. ^1^H- and ^13^CNMR data were in accordance with those reported in the literature [50]; IR (cm^−1^): 3350, 2987, 1698, 1659, 1506, 1450, 1370, 1299, 1084, 836, 762, 693; ESI-TOF HR: *m*/*z* calculated for C_19_H_17_FN_2_O_3_: 363.1772 [M+Na]^+^, found 363.1775; elemental analysis calculated for C_19_H_17_FN_2_O_3_: C, 67.05%; H, 5.03%; N, 8.23%, found C 67.03%; H, 5.06%; N, 8.26%. 

### 4.7. Product ***4h***: 4-(4-fluorophenyl)-1,2,3,4-tetrahydro-6-methyl-2-oxo-N-phenylpyrimi-dinecarboxamide

The crude product was purified by crystallization from ethanol to obtain a suitable product in the form of white crystals; mp. 218–219 °C; ^1^H NMR (400 MHz, DMSO-*d*_6_) δ 9.51 (s, 1H), 8.71 (d, *J* = 2.0 Hz, 1H), 7.58–7.49 (m, 3H), 7.32–7.26 (m, 2H), 7.22 (dd, *J* = 8.5, 7.3 Hz, 2H), 7.17–7.10 (m, 2H), 7.01–6.95 (m, 1H), 5.38 (d, *J* = 2.9 Hz, 1H), 2.03 (s, 3H); ^13^C NMR (100 MHz, DMSO-*d*_6_) δ 165.7, 152.9, 140.9, 139.6, 139.02 128.9, 128.8, 128.7, 123.5, 120.1, 115.7, 115.5, 105.7, 54.9, 31.1, 17.5; ^19^F NMR (376 MHz, DMSO-*d*_6_) δ −115.4. ^1^H- and ^13^CNMR data were in accordance with those reported in the literature [51]; IR (cm^−1^): 3277, 2986, 1702, 1675, 1629, 1509, 1439, 1321, 1244, 1096, 841, 755, 692; ESI-TOF HR: *m*/*z* calculated for C_18_H_16_FN_3_O_2_: 326.0787 [M+H]^+^, found 326.0791; elemental analysis calculated for C_18_H_16_FN_3_O_2_: C, 66.45%; H, 4.96%; N, 12.92%, found C, 66.28%; H, 4.52%; N, 13.01%.

### 4.8. Product ***4i***: 4-(4-fluorophenyl)-1,2,3,4-tetrahydro-6-methyl-2-oxopyrimidine-5-carbo-xamide

The crude product was purified by crystallization from ethanol to obtain a suitable product in the form of white crystals; mp. 197–198 °C; ^1^H NMR (400 MHz, DMSO-*d*_6_) δ 8.58 (d, *J* = 2.0 Hz, 1H), 7.49 (d, *J* = 2.4 Hz, 1H), 7.27 (dd, *J* = 8.6, 5.6 Hz, 2H), 7.12 (t, *J* = 8.9 Hz, 2H), 6.88 (s, 2H), 5.22 (d, *J* = 3.0 Hz, 1H), 2.05 (s, 3H); ^13^C NMR (100 MHz, DMSO-*d*_6_) δ 168.5, 153.0, 141.0, 141.0, 139.4, 128.9, 128.8, 115.5, 115.3, 104.7, 54.5, 17.5; ^19^F NMR (376 MHz, DMSO-*d*_6_) δ −115.6; IR (cm^−1^): 3243, 2936, 1700, 1682, 1636, 1507, 1457, 1214, 1094, 836, 798, 647; elemental analysis calculated for C_12_H_12_FN_3_O_2_: C, 58.06%; H, 4.67%; N, 16.93%, found C, 58.03%; H, 4.65%; N, 16.95%.

### 4.9. Product ***5a***: Ethyl 4-(4-fluorophenyl)-6-methyl-1,2,3,4-tetrahydro-2-thioxopyrimi-dine-5-carboxylate

The crude product was purified by crystallization from ethanol to obtain a suitable product in the form of yellow crystals; mp. 191–192 °C [Lit mp. 190–192 °C] [52]; ^1^H NMR (400 MHz, DMSO-*d*_6_) δ 10.31 (s, 1H), 9.61 (s, 1H), 7.39–7.09 (m, 4H), 5.16 (d, *J* = 3.7 Hz, 1H), 3.99 (dd, *J* = 7.1, 2.4 Hz, 2H), 2.28 (s, 3H), 1.08 (t, *J* = 7.1 Hz, 3H); ^13^C NMR (100 MHz, DMSO-*d*_6_) δ 174.3, 165.7, 152.4, 148.9, 141.6, 128.7, 128.6, 115.6, 115.4, 99.6, 59.6, 53.8, 18.2, 14.5; ^19^F NMR (376 MHz, DMSO-*d*_6_) δ −114.7. ^1^H- and ^13^CNMR data were in accordance with those reported in the literature [53]; IR (cm^−1^): 3326, 2985, 1682, 1603, 1574, 1467, 1334, 1283, 1176, 1118, 1029, 854, 759, 649; ESI-TOF HR: *m*/*z* calculated for C_14_H_15_FN_2_O_2_S: 295.0916 [M+H]^+^, found 295.0947; elemental analysis calculated for C_14_H_15_FN_2_O_2_S: C, 57.13%; H, 5.14%; N, 9.52%; S, 10.89%, found C, 57.17%; H, 5.13%; N, 9.58%; S, 10.79%. 

### 4.10. Product ***5b***: Methyl 4-(4-fluorophenyl)-6-methyl-1,2,3,4-tetrahydro-2-thioxopyrimi-dine-5-carboxylate

The crude product was purified by crystallization from ethanol to obtain a suitable product in the form of yellow crystals; mp. 189–190 °C [Lit mp. 191–193 °C] [54], ^1^H NMR (400 MHz, DMSO-*d*_6_) δ 10.34 (s, 1H), 9.63 (dd, *J* = 3.8, 1.9 Hz, 1H), 7.35–7.09 (m, 4H), 5.16 (d, *J* = 3.8 Hz, 1H), 3.54 (s, 3H), 2.28 (s, 3H); ^13^C NMR (100 MHz, DMSO-*d*_6_) δ 174.7, 166.0, 163.2, 160.8, 145.9, 140.0, 140.01, 128.8, 128.7, 115.9, 115.7, 100.8, 53.7, 51.5, 17.6; ^19^F NMR (376 MHz, DMSO-*d*_6_) δ −114.6. ^1^H- and ^13^CNMR data were in accordance with those reported in the literature. [55]; IR (cm^−1^): 3309, 2998, 1669, 1574, 1470, 1385, 1200, 1117, 1094, 838, 767, 651; ESI-TOF HR: *m*/*z* calculated for C_13_H_13_FN_2_O_2_S: 303.0579 [M+Na]^+^, found 303.0572; elemental analysis calculated for C_13_H_13_FN_2_O_2_S: C, 55.70%; H, 4.67%; N, 9.99%; S, 11.44%, found C, 55.74%; H, 4.65%; N, 10.09%; S, 11.41%.

### 4.11. Product ***5e***: Ethyl 4-(4-fluorophenyl)-6-phenyl-1,2,3,4-tetrahydro-2-thioxopyrimi-dine-5-carboxylate

The crude product was purified by crystallization from ethanol to obtain a suitable product in the form of yellow crystals; mp. 204–205 °C; 1H NMR (400 MHz, DMSO-d6) δ 9.73 (s, 1H), 7.43–7.17 (m, 9H), 5.26 (s, 1H), 3.73 (q, J = 7.1 Hz, 2H), 0.71 (t, J = 7.1 Hz, 3H); 13C NMR (100 MHz, DMSO-d6) δ 174.9, 165.3, 146.3, 139.7, 134.3, 129.6, 129.1, 128.9, 128.8, 128.16, 116.0, 115.8, 102.2, 59., 53.9, 13.7; 19F NMR (376 MHz, DMSO-d6) δ −114.5. 1H- and 13CNMR data were in accordance with those reported in the literature [56]; IR (cm^−1^): 3297, 2983, 1672, 1570, 1464, 1333, 1201, 1135, 1118, 1029, 835, 769, 693; ESI-TOF HR: m/z calculated for C19H17FN2O2S: 713.2068 [2M+H]+, found 713.2074; elemental analysis calculated for C19H17FN2O2S: C, 64.03%; H, 4.81%; N, 7.86%; S, 9.00%, found C, 64.05%; H, 4.84%; N, 7.91%; S, 9.06%.

### 4.12. Product ***5g***: Ethyl 4-(4-fluorophenyl)-6-(3-methylphenyl)-1,2,3,4-tetrahydro-2-thi-oxopyrimi-dine-5-carboxylate

The crude product was purified by crystallization from ethanol to obtain a suitable product in the form of yellow crystals; mp. 196–197 °C; ^1^H NMR (400 MHz, DMSO-*d*_6_) δ 9.71 (s, 1H), 7.46–7.33 (m, 2H), 7.33–7.19 (m, 4H), 7.19–6.89 (m, 3H), 5.24 (s, 1H), 3.74 (qd, *J* = 7.0, 2.6 Hz, 2H), 2.31 (s, 3H), 0.73 (t, *J* = 7.1 Hz, 3H); ^13^C NMR (100 MHz, DMSO-*d*_6_) δ 175.0, 165.3, 146.4, 137.2, 134.2, 130.2, 129.5, 128.9, 128.8, 128.0, 126.3, 116.0, 115.8, 102.0, 59.9, 53.9, 21.2, 13.7; ^19^F NMR (376 MHz, DMSO-*d*_6_) δ −114.5; IR (cm^−1^): 3304, 2979, 1674, 1567, 1462, 1332, 1181, 1132, 1095, 856, 768, 698; ESI-TOF HR: *m*/*z* calculated for C_20_H_19_FN_2_O_2_: 741.2381 [2M+H]^+^, found 741.2386; elemental analysis calculated for C_20_H_19_FN_2_O_2_: C, 64.85%; H, 5.17%; N, 7.56%; S, 8.66%, found C, 64.83%; H, 5.11%; N, 7.51%; S, 8.68%.

### 4.13. Product ***5h***: Ethyl 4-(4-fluorophenyl)-6-(4-methylphenyl)-1,2,3,4-tetrahydro-2-thioxo-pyrimidine-5-carboxylate

The crude product was purified by crystallization from ethanol to obtain a suitable product in the form of yellow crystals; mp. 198–199 °C; ^1^H NMR (400 MHz, DMSO-*d*_6_) δ 9.72 (s, 1H), 7.38 (dd, *J* = 8.6, 5.6 Hz, 2H), 7.28–7.14 (m, 6H), 5.24 (s, 1H), 3.75 (q, *J* = 7.1 Hz, 2H), 2.33 (s, 3H), 0.75 (t, *J* = 7.1 Hz, 3H); ^13^C NMR (100 MHz, DMSO-*d*_6_) δ 175.0, 165.3, 146.4, 139.2, 131.3, 129.1, 128.9, 128.8, 128.6, 116.0, 115.8, 101.9, 59.9, 53.8, 21.4, 13.8; ^19^F NMR (376 MHz, DMSO-*d*_6_) δ −114.6; IR (cm^−1^): 3292, 2993, 1668, 1570, 1455, 1334, 1201, 1135, 1051, 821, 754, 641; ESI-TOF HR: *m*/*z* calculated for C_20_H_19_FN_2_O_2_S: 741.2381 [2M+H]^+^, found 741.2418; elemental analysis calculated for C_20_H_19_FN_2_O_2_S: C, 64.85%; H, 5.17%; N, 7.56%; S, 8.66%, found C, 64.83%; H, 5.18%; N, 7.55%; S, 8.68%. 

### 4.14. Product ***5j***: 4-(4-fluorophenyl)-1,2,3,4-tetrahydro-6-methyl-N-phenyl-2-thioxo-pyrimidine-5-carboxamide

The crude product was purified by crystallization from ethanol to obtain a suitable product in the form of yellow crystals; mp. 182–183 °C; ^1^H NMR (400 MHz, DMSO-*d*_6_) δ 9.98 (s, 1H), 9.69 (s, 1H), 9.40 (d, *J* = 2.8 Hz, 1H), 7.59–7.45 (m, 2H), 7.33–7.11 (m, 6H), 7.04–6.94 (m, 1H), 5.46–5.28 (m, 1H), 2.05 (s, 3H); ^13^C NMR (101 MHz, DMSO-*d*_6_) δ 174.5, 165.3, 139.3, 136.1, 129.0, 128.9, 128.9, 123.8, 120.1, 115.9, 115.7, 107.5, 54.9, 16.9; ^19^F NMR (376 MHz, DMSO-*d*_6_) δ −114.7. ^1^H- and ^13^CNMR data were in accordance with those reported in the literature. [57] IR (cm^−1^): 3262, 2951, 1682, 1631, 1585, 1484, 1338, 1208, 1162, 842, 750, 689; ESI-TOF HR: *m*/*z* calculated for C_18_H_16_FN_3_OS: 364.2380 [M+Na]^+^, found 364.2382; elemental analysis calculated for C_18_H_16_FN_3_OS: C, 63.32%; H, 4.72%; N, 12.31%; S, 9.39%, found C, 63.29%; H, 4.75%; N, 12.30%; S, 9.41%.

### 4.15. Product ***5k***: 4-(4-fluorophenyl)-1,2,3,4-tetrahydro-6-methyl-2-thioxopyrimidine-5-carboxamide

The crude product was purified by crystallization from ethanol to obtain a suitable product in the form of yellow crystals; mp. 138–139 °C; ^1^H NMR (400 MHz, DMSO-*d*_6_) δ 9.83 (d, *J* = 1.9 Hz, 1H), 9.33 (d, *J* = 1.5 Hz, 1H), 7.25 (dd, *J* = 8.6, 5.7 Hz, 2H), 7.15 (t, *J* = 8.9 Hz, 2H), 5.25 (d, *J* = 3.4 Hz, 1H), 2.06 (s, 3H); ^13^C NMR (100 MHz, DMSO-*d*_6_) δ 174.3, 168.2, 139.7, 136.4, 129.0, 129.0, 115.7, 115.5, 106.6, 54.5, 31.1, 16.9; ^19^F NMR (376 MHz, DMSO-*d*_6_) δ −114.9; IR (cm^−1^): 3370, 2983, 1681, 1609, 1506, 1479, 1210, 1186, 1097, 841, 687, 597; ESI-TOF HR: *m*/*z* calculated for C_12_H_12_FN_3_OS: 288.1402 [M+Na]^+^, found 288.1402; elemental analysis calculated for C_12_H_12_FN_3_OS: C, 54.32%; H, 4.56%; N, 15.84%; S, 12.09%, found C, 54.33%; H, 4.58%; N, 15.82%; S, 12.11%.

### 4.16. Product ***6a***: Ethyl 1,2,3,4-tetrahydro-6-methyl-2-oxo-4-phenylpyrimidine-5-carboxy-late

The crude product was purified by crystallization from ethanol to obtain a suitable product in the form of yellow crystals; mp. 209–210 °C [Lit mp. 211–213 °C] [58]; ^1^H NMR (400 MHz, DMSO-*d*_6_) δ 9.18–9.06 (m, 1H), 7.68 (d, *J* = 2.8 Hz, 1H), 7.34–7.13 (m, 5H), 5.13 (d, *J* = 3.4 Hz, 1H), 3.96 (dd, *J* = 7.0, 1.1 Hz, 2H), 2.23 (s, 3H), 1.07 (t, *J* = 7.1 Hz, 3H); ^13^C NMR (100 MHz, DMSO-*d*_6_) δ 165.8, 148.7, 145.3, 128.8, 127.7, 126.6, 99.7, 59.6, 54.4, 18.2, 14.5. ^1^H- and ^13^CNMR data were in accordance with those reported in the literature [59]. IR (cm^−1^): 3245, 2978, 1700, 1647, 1465, 1313, 1220, 1090, 817, 782, 698; ESI-TOF HR: *m*/*z* calculated for C_14_H_16_N_2_O_3_: 261.0302 [M+H]^+^, found 261.0303; elemental analysis calculated for C_14_H_16_N_2_O_3_: C, 64.60%; H, 6.20%; N, 10.76%, found C, 64.65%; H, 6.17%; N, 10.79%.

### 4.17. Product ***6b***: Ethyl 1,2,3,4-tetrahydro-6-methyl-2-thioxo-4-phenylpyrimidine-5-carbo-xylate

The crude product was purified by crystallization from ethanol to obtain a suitable product in the form of yellow crystals; mp. 214–215 °C [Lit mp. 211–212 °C] [60], ^1^H NMR (400 MHz, DMSO-*d*_6_) δ 10.28 (s, 1H), 9.60 (s, 1H), 7.42–7.16 (m, 5H), 5.16 (d, *J* = 3.8 Hz, 1H), 3.99 (q, *J* = 7.0 Hz, 2H), 2.27 (s, 3H), 1.08 (t, *J* = 7.1 Hz, 3H); ^13^C NMR (100 MHz, DMSO-*d*_6_) δ 174.7, 165.5, 145.4, 143.9, 129.0, 128.1, 126.8, 101.2, 60.0, 54.5, 17.6, 14.4. ^1^H- and ^13^CNMR data were in accordance with those reported in the literature [61], IR (cm^−1^): 3328, 2978, 1670, 1574, 1465, 1370; 1283, 1176, 1118, 1027, 833, 760, 692; ESI-TOF HR: *m*/*z* calculated for C_14_H_16_N_2_O_2_S: 277.0417 [M+H]^+^, found 277.0419; elemental analysis calculated C_14_H_16_N_2_O_2_S: C, 60.86%; H, 5.84%; N, 10.14%; S, 11.61%, found C, 60.81%; H, 5.85%; N, 10.15%; S, 11.63%. 

## 5. Results and Discussion

### 5.1. Chemistry

Recently, we showed that *δ*-lactones show high biological activity in terms of action on bacterial lipopolysaccharide (LPS) in model strains of *Escherichia coli* K12 (without LPS in its structure) and R2–R4 (LPSs of different lengths in its structure) [41]. Therefore, we were interested in whether the same effect of fluorine presented in the structure of the investigated DHPMs would modulate their antimicrobial activity, as was observed for δ-lactones. For this purpose, it was necessary to develop a method to provides DHPMs containing fluorine in their structure. A series of target DHPM 4s were prepared via Biginelli reaction and catalyzed by ammonium chloride under solvent-free conditions (Figure 2). The desired DHPMs were obtained with the yields ranging from 56% to 82% (Table 1). The developed method turned out to also be effective for the synthesis of thioxopyrimidine derivative **5j**, which was provided with up to 76% of isolated yield. In order to verify the impact of the fluorine atom on biological activity, DHPMs without fluorine substituents (**6a** and **6b**) were obtained (Table 1). 

The structures of all obtained compounds were confirmed using NMR and mass spectroscopy. The analytical data of all synthesized DHPMs are presented in the Experimental Section. 

### 5.2. Cytotoxic Studies of the Library of Peptidomimetics

Analyzed DHPMs had an inhibitory effect on each studied bacterial model. Varied inhibitory activities were noted, depending on the nature of the substituent R_1_ in the pyrimidine ring of tested compounds **4** and **5**. Interestingly, DHPM derivatives **4b**, **4c**, **4h**, **4i**, **5b**, **5g**, **5h**, **5j** and **5k** exhibited selectivity towards the K12 and R2 strains. The presence of more lipophilic groups at the R_1_ position of tested compounds **4d** and **4e** resulted in increased inhibitory activity against the R3 strain. It is clearly visible that the structure of the tested compounds is closely related to their inhibitory activity. The R3 and R4 strains turned out to be less sensitive to tested compounds. The 4-fluorine group introduced into the aromatic ring did not have such a pronounced effect on the increase in activity of the tested compounds, as was the case of the *δ*-lactones tested previously [41]. MIC values for each model *E. coli* R2–R4 and K12 strains were visible on all analyzed microplates after the addition of the microbial growth index (resazurin)(Table 2).

In all analyzed plates where model strains K12 and R2–R4 of *E. coli* bacteria were treated with the analyzed compounds, MIC and MBC test values were observed for all 16 compounds but at different levels. Bacterial strains R3 and R4 were more susceptible to modification with these compounds (visible dilutions of 10–2, corresponding to concentrations of 0.0025 µM) than strains K12 and R2 (visible dilutions of 10–6, corresponding to concentrations of 0.2 µM). Strain R4 was the most sensitive of all strains, possibly due to having the longest lipopolysaccharide chain length. Based on the values in both types of tests and after the color-change analysis, the compounds **4b**, **4c**, **4h**, **4i**, **5b**, **5g**, **5h**, **5j** and **5k** were used for further analyses. In all analyzed cases, MBC values were approximately 40 times higher than MIC values (Figure 3). Modification of functional groups in the tested compounds significantly changed the MBC/MIC ratios, (Figure 3, Figure 4 and Figure 5 and Table 1). 

### 5.3. Analysis of Bacterial DNA Isolated from E. coli R2–R4 Strains Modified with 3,4-dihydropyrimidin-2(1H)-ones (DHPMs)

The obtained MIC values, as well as those obtained in our previous studies with the various types of analyzed compounds [37,38,39,40,41,42], indicate that 3,4-dihydropyrimidin-2(1H)-ones (DHPMs) also have a toxic effect on the analyzed model bacterial strains K12 and R2–R4 due to the type of substituent in the aromatic rings and the appropriate length of the alkyl chain with the specified type of substituent. Selected compounds **4b**, **4c**, **4h**, **4i**, **5b**, **5g**, **5h**, **5j** and **5k** were used to modify model *E. coli* strains and digest them with Fpg protein from the group of repair glycosylases, which is a marker of oxidative stress [37,38,39,40,41,42]. We wanted to observe the effect of modification on the size and location of the oxidative damage (base type) in the DNA chain, which is a substantial reduction in the three forms of bacterial DNA: ccc, oc and linear form, as observed in previous studies [37,38,39,40,41,42]. The results of bacterial DNA modified with 3,4-dihydropyrimidin-2(1H)-ones (DHPMs) (Figure 6; with the action of Fpg) showed that all analyzed compounds with different alkyl chain lengths and substituents containing a hydroxyl group or an amino group together with substituents containing bromine and chlorine can strongly change the topology of the plasmid form, even after digestion with Fpg protein (Appendix A) and are highly toxic to it, similar to the observations of previous studies [37,38,39,40,41,42]. The obtained results were also statistically significant at the level of *p* < 0.05. (Figure 6).

The obtained results indicate that these compounds can also potentially be used as “substitutes for” commonly used antibiotics (Figure 7 and Figure 8, Appendix A).

Large modifications of plasmid DNA were observed for compounds **4b**, **4c**, **4h**, **4i**, **5b**, **5g**, **5h**, **5j** and **5k** among all analyzed compounds. Modifications with antibiotics were smaller and not as clear as in the case of the analyzed DHPMs. The sensitivity of *E. coli* strains to the cytotoxic effect of the compounds used and after Fpg protein digestion was as follows: R4 > R2 > R3 > K12; this effect was very similar to that observed in our previous studies [37,38,39,40,41,42]. This indicates a very high cytotoxicity of the analyzed DHPMs towards bacterial DNA, probably resulting from the modification of the components of the bacterial membrane and the LPS contained in it, which may induce specific enzymes from the group of topoisomerases and helicases, destabilizing the structure of the exposed DNA bases. The stabilization of the complex that regulates these enzymes is perhaps necessary for cell survival. Blocking these enzymes inhibits DNA replication and rewriting, which can affect its total amount.

## 6. Conclusions

The analysis of the presented compounds through a complex cycle of their synthesis reactions may constitute a potential source of innovative, cheap substitutes for antibiotics against various types of bacterial microorganisms (LPS). We focused on the structure–activity relationship of compounds with the 3,4-dihydropyrimidin-2(1H)-one scaffold. The obtained results show a strong influence of the activity of all 16 analyzed compounds on the values of MIC and MBC, as well as MBC/MIC for various strains of *E. coli*: R2–R4 and K12. Based on the analysis of the above studies, nine compounds (**4b**, **4c**, **4h**, **4i**, **5b**, **5g**, **5h**, **5j** and **5k**) were selected for further research. The repair activity using the Fpg-glycosylase protein of the BER pathway (base excision repair) was then compared, which is in accordance with our research hypothesis. The above results are very important for research on the mechanism of cytotoxic action of new compounds as innovative and safe drugs based on 3,4-dihydropyrimidin-2(1H)-one (DHPM) derivatives, which may lead to the destruction of the bacterial cell membrane by changing its surface charge and may play an important role in changing its electrokinetic potential, expressing the reversal of burdens. A particular effect was observed for the mentioned nine select compounds, which showed certain MIC values and MBC/MIC ratios. Compounds **5b**, **5j** and **5k** showed superselectivity in all analyzed bacterial strains, even differentiating cytotoxicity in the K12 strain. The described compounds can be highly specific for pathogenic *E. coli* strains on the basis of the model strains used. In the future, cytotoxicity studies should also be carried out using various cell lines and cultures to assess the biocompatibility of the tested compounds with active peptidomimetics. 

## Figures and Tables

**Figure 1 membranes-12-00238-f001:**
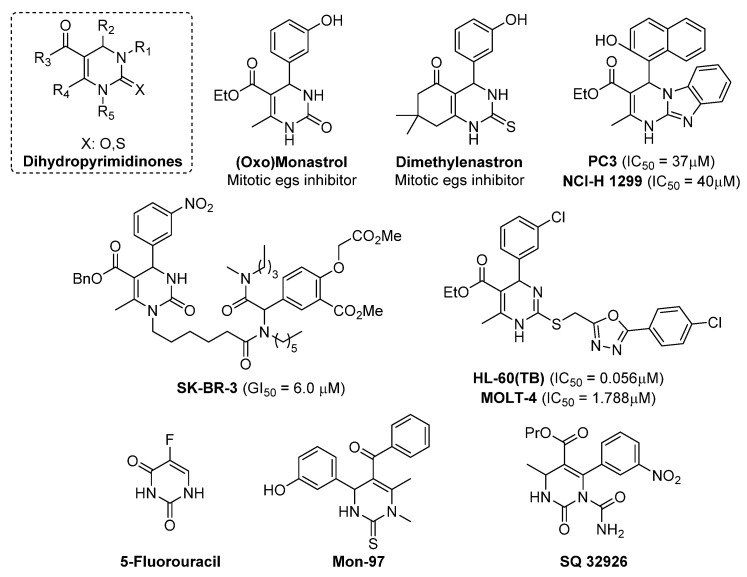
Pharmaceutically relevant 3,4-Dihydropyrimidin-2(1H)-ones (DHPMs) based on [1,2,3].

**Figure 2 membranes-12-00238-f002:**
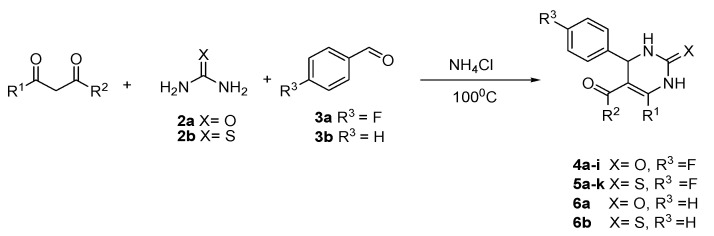
Synthesis of the studied DHPMs.

**Figure 3 membranes-12-00238-f003:**
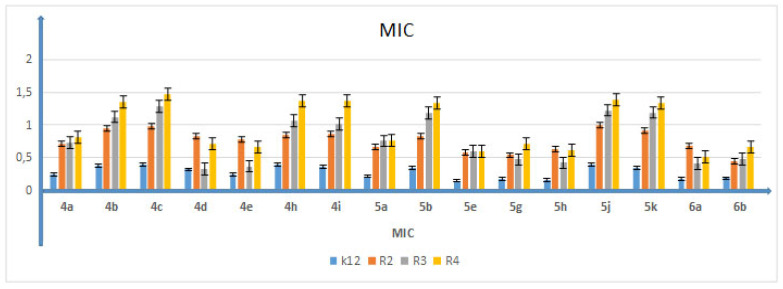
Minimum inhibitory concentration (MIC) of 3,4-dihydropyrimidin-2(1H)-ones (DHPMs) in model bacterial strains. The *x*-axis features 3,4-dihydropyrimidin-2(1H)-ones (DHPMs) used sequentially. The *y*-axis shows the MIC value in µg/mL^−1^. Investigated strains of *E. coli*: K12 as control (blue), R2 strain (orange), R3 strain (gray) and R4 strain (yellow). The *y*-axis shows the MBC value in µg/mL^−1^. The order in which the compounds were applied to the plate is shown in Appendix A.

**Figure 4 membranes-12-00238-f004:**
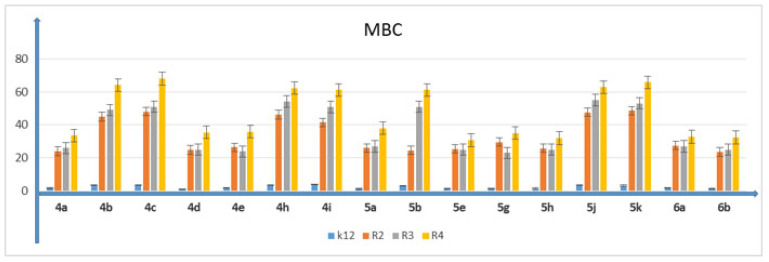
Minimum bactericidal concentration (MBC) of 3,4-dihydropyrimidin-2(1H)-ones (DHPMs) in model bacterial strains. The *x*-axis features 3,4-dihydropyrimidin-2(1H)-ones (DHPMs) used sequentially. The *y*-axis shows the MIC value in µg/mL^−1^. Investigated strains of *E. coli*: K12 as control (blue), R2 strain (orange), R3 strain (gray) and R4 strain (yellow). The *y*-axis shows the MBC value in µg/mL^−1^. The order in which the compounds were applied to the plate is shown in Appendix A.

**Figure 5 membranes-12-00238-f005:**
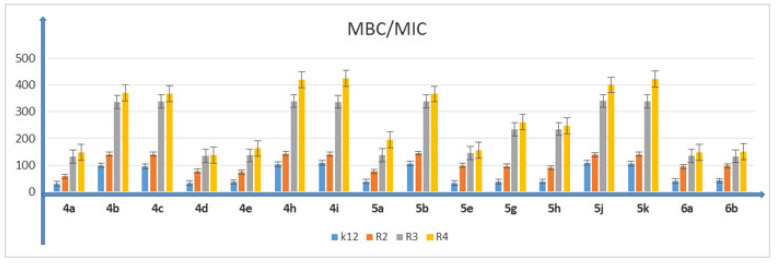
The ratio of MBC/MIC of 3,4-dihydropyrimidin-2(1H)-ones (DHPMs) in model bacterial strains. The *x*-axis features 3,4-dihydropyrimidin-2(1H)-ones (DHPMs) used sequentially. The *y*-axis shows the MIC value in µg/mL^−1^. Investigated strains of *E. coli*: K12 as control (blue), R2 strain (orange), R3 strain (gray) and R4 strain (yellow). The *y*-axis shows the MBC value in µg/mL^−1^. The order in which the compounds were applied to the plate is shown in Appendix A.

**Figure 6 membranes-12-00238-f006:**
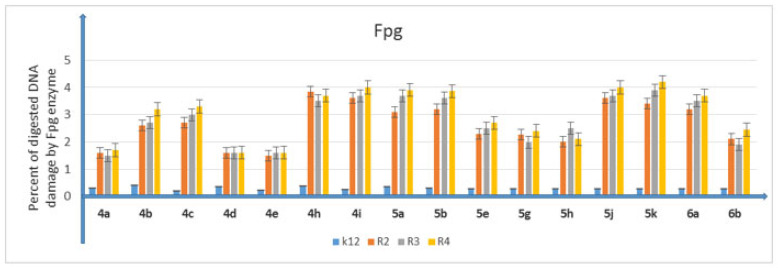
Percentage of plasmid DNA recognized by Fpg enzyme (*y*-axis) with model bacterial strains K12 and R2–R4 (*x*-axis). Compounds **4b**, **4c**, **4h**, **4i**, **5b**, **5g**, **5h**, **5j** and **5k** were statistically significant at * *p* < 0.05 (see Table 1).

**Figure 7 membranes-12-00238-f007:**
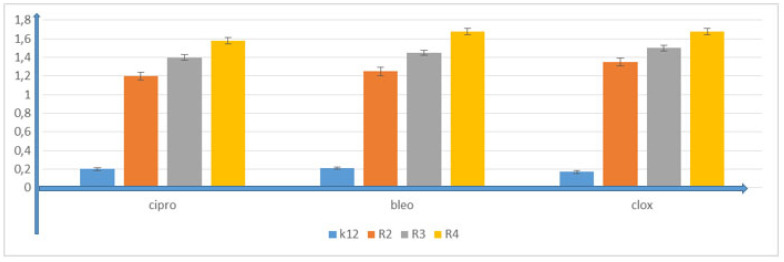
Examples of MIC with model bacterial strains K12, R2, R3 and R4 for study of the mechanisms of antibiotics ciprofloxacin and bleomycin. The *x*-axis features antibiotics used sequentially. The *y*-axis features the MIC value in µg/mL^−1^.

**Figure 8 membranes-12-00238-f008:**
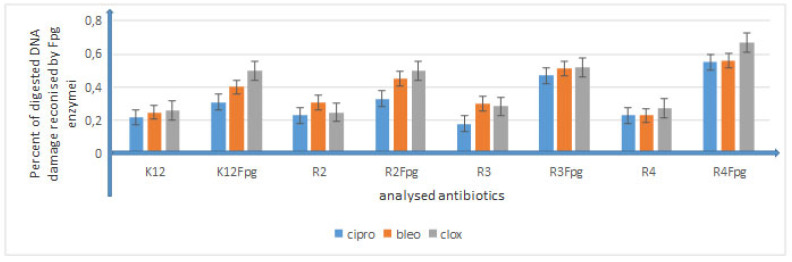
Percentage of bacterial DNA recognized by Fpg enzyme in model bacterial strains after ciprofloxacin, bleomycin and cloxacillin treatment. The compounds were statistically significant at *p* < 0.05.

**Table 1 membranes-12-00238-t001:** Yield of DHPMs **4**, **5** and **6**.

Entry	Compound	X	R^1^	R^2^	R^3^	Yield
1	**4a**	O	Me	OEt	F	67%
2	**4b**	O	Me	OMe	F	56%
3	**4c**	O	Me	O*i*-Bu	F	63%
4	**4d**	O	Pr	OEt	F	69%
5	**4e**	O	Ph	OEt	F	82%
6	**4h**	O	Me	NHPh	F	67%
7	**4i**	O	Me	NH_2_	F	57%
8	**5a**	S	Me	OEt	F	54%
9	**5b**	S	Me	OMe	F	55%
10	**5e**	S	Ph	OEt	F	54%
11	**5g**	S	Ph(3-Me)	OEt	F	58%
12	**5h**	S	Ph(4-Me)	OEt	F	63%
13	**5j**	S	Me	NHPh	F	76%
14	**5k**	S	Me	NH_2_	F	61%
15	**6a**	O	Me	OEt	H	70%
16	**6b**	S	Me	OEt	H	59%

**Table 2 membranes-12-00238-t002:** Statistical analysis of analyzed compounds by MIC, MBC and MBC/MIC; * *p* < 0.05, ** *p* < 0.01 and *** *p* < 0.001.

No. of Samples	4b	4c	4h	4i	5b,5g	5h	5j,5k	Type of Test
K12	*	*	*	**	**	*	***	MIC
R2	*	*	*	**	**	*	***	MIC
R3	*	*	*	**	**	*	***	MIC
R4	*	*	*	**	**	*	***	MIC
K12	*	*	**	*	**	**	**	MBC
R2	**	*	**	*	**	**	**	MBC
R3	**	*	**	*	**	**	**	MBC
R4	**	*	**	*	**	**	**	MBC
K12	*	**	*	*	*	**	***	MBC/MIC
R2	*	**	*	*	*	**	***	MBC/MIC
R3	*	**	*	*	*	**	***	MBC/MIC
R4	*	**	*	*	*	**	***	MBC/MIC

## Data Availability

The data presented in this study are available on request from the corresponding author.

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
