# Peer review of "The Evaluation of DHPMs as Biotoxic Agents on Pathogen Bacterial Membranes"

_membranes, 2022, doi:10.3390/membranes12020238_

Round 1

Reviewer 1 Report

Manuscript is well written, however, minor changes are suggested. After incorporation of minor changes, the manuscript can be accepted for published in membranes.

Suggestions/comments:

  1. There are numerous formatting issues in the whole manuscript, specially heading in the experimental section.
  2. Experimental section should be combined with materials and methods sections.
  3. Y axis and X axis must be properly labelled. Labelling is missing in some graphs.

Author Response

Thank you very much for all the valuable comments that contributed to the improvement of the quality of the manuscript

Reviewer 1

Suggestions/comments:

  1. There are numerous formatting issues in the whole manuscript, specially heading in the experimental section.

The entire article was formatted with particular regard to the experimental part as suggested

  1. Experimental section should be combined with materials and methods sections.

We followed the template determined by the journal

  1. Y axis and X axis must be properly labelled. Labelling is missing in some graphs.

The Y axis and the X axis have been marked appropriately by adding arrows. The legend has been included in all charts

Reviewer 2 Report

The authors discuss the synthesis of 3,4-dihydropyrimidin-2(1H)-ones (DHPMs) and further investigate their biological activity against pathogenic E. coli strains through preliminary studies including MIC and MBC tests and digestion of Fpg after modification of bacterial DNA. These results suggest that these compounds have potential as new antibacterial agents. Generally, this article has clear logic and some interesting results. But there also exist some defects that need to be improved.

  1. Line 47-51, some background information seems omitted but really necessary, such as the role of LPS in pathogenic E. coli and its relationship with oxidative damage. Also, the authors should highlight the risk of pathogenic E. coli in clinical blood infections.

  1. Line 51-52, ‘There is still a need for further in vivo studies to better delineate the pharmacological potential of this class of substances. Here, we present an attempt to answer this question.’ No any in vivo studies were presented in this manuscript. Therefore, this statement is not unreasonable.

  1. Line 315-339, The protocols for MIC, MBC and Fpg digestion experiments are not clear, how to calculate the percent of digested DNA damage by Fpg enzyme.

  1. Line 345-346, ‘E. coli K12 (without LPS in its structure)’, please confirm this information.

  1. Line 427-428, Strain R4 was the most sensitive of all strains? How to obtain this result. According to Figure 3 and 4, these compounds showed the weakest activity against strain R4. Additionally, author stated that the longest lipopolysaccharide chain length of strain R4 makes it the most sensitive of all strains, please give more explanations for this.

  1. Line 446-451, The results showed that all analyzed compounds with different alkyl chain length and substituents containing a hydroxyl group or an amino group together with substituents containing bromine and chlorine can strongly change the topology of the plasmid form, even after digestion with Fpg protein. But why just fluorine group but not bromine and chlorine group were introduced into aromatic ring to evaluate the activity of the tested compounds. The authors can also further explore the role of bromine and chlorine group in the cellular reaction.

  1. Line 489-493, The cytotoxicity of DHPMs to bacterial DNA results from the modification of the components of the bacterial membrane and the LPS contained in it? It is hard to understand this sentence. Whether these compounds act by disrupting bacterial membrane? Please give experimental evidence for this. In addition, the relationship between DNA damage and membrane disruption is unclear.

  1. Supplementary Materials for this manuscript are missing, please upload it.

  1. There are some grammatical errors in the context and figures. For example,

Line 40: remove ‘one’

Line 438: the position of ‘also’ seems insuitable

Line 442: please correct the form of ‘E.coli

Line 456: ‘similarly’ should be substituted for ‘similar’

Line 518: please substitute ‘test’ for ‘tested’

Figure 8: please remove the ‘i’ after ‘enzyme’ and ‘antibiotics’

Author Response

Thank you very much for all the valuable comments that contributed to the improvement of the quality of the manuscript

Reviewer 2

The authors discuss the synthesis of 3,4-dihydropyrimidin-2(1H)-ones (DHPMs) and further investigate their biological activity against pathogenic E. coli strains through preliminary studies including MIC and MBC tests and digestion of Fpg after modification of bacterial DNA. These results suggest that these compounds have potential as new antibacterial agents. Generally, this article has clear logic and some interesting results. But there also exist some defects that need to be improved.

  1. Line 47-51, some background information seems omitted but really necessary, such as the role of LPS in pathogenic E. coli and its relationship with oxidative damage. Also, the authors should highlight the risk of pathogenic E. coli in clinical blood infections.

 All these symptoms have been described very precisely in the cited publications, therefore we did not want to duplicate them in the descriptions

44.Maciejewska, A.; Kaszowska, M.; Jachymek, W.; Lugowski, C.; Lukasiewicz, J. Lipopolysaccharide‐linked Enterobacterial Common Antigen (ECALPS) Occurs in Rough Strains of Escherichia coli R1, R2, and R4. Int. J. Mol. Sci. 2020, 21, 6038. doi: 10.3390/ijms21176038.

45.Prost, M.E.; Prost, R. Basic parameters of evaluation of the effectiveness of antibiotic therapy. OphthaTherapy 2017, 4, 233–236. doi: 10.24292/01.ot.291217.06.

  1. Line 51-52, ‘There is still a need for further in vivo studies to better delineate the pharmacological potential of this class of substances. Here, we present an attempt to answer this question.’ No any in vivo studies were presented in this manuscript. Therefore, this statement is not unreasonable.

The basic research aimed at checking whether the tested compounds show activity against selected strains of bacteria was presented. The obtained results confirmed the activity on selected strains, which was the aim of the conducted research. Further research is being continued and subsequent aspects, incl. those related to cytotoxicity will be the subject of further work. example are given in the Literature section.

38.Kowalczyk, P.; Madej, A.; Szymczak, M.; Ostaszewski, R. α‐Amidoamids as New Replacements of Antibiotics—Research on the Chosen K12, R2–R4 E. coli Strains. Materials 2020, 13, 5169. doi: 10.3390/ma13225169.

39.Kowalczyk, P.; Borkowski, A.; Czerwonka, G.; Cłapa, T.; Cieśla, J.; Misiewicz, A.; Borowiec, M.; Szala, M. The microbial toxicity of quaternary ammonium ionic liquids is dependent on the type of lipopolysaccharide. J. Mol. Liq. 2018, 266, 540–547. doi: 10.1016/j.molliq.2018.06.102.

40.Borkowski, A.; Kowalczyk, P.; Czerwonka, G.; Cieśla, J.; Cłapa, T.; Misiewicz, A.; Szala, M.; Drabik, M. Interaction of quaternary ammonium ionic liquids with bacterial membranes—Studies with Escherichia coli R1–R4‐type lipopolysaccharides. J. Mol. Liq. 2017, 246, 282–289. doi: 10.1016/j.molliq.2017.09.074.

41.Kowalczyk, P.; Gawdzik, B.; Trzepizur, D.; Szymczak, M.; Skiba, G.; Raj, S.; Kramkowski, K.; Lizut, R.; Ostaszewski, R. δ‐Lactones—A New Class of Compounds That Are Toxic to E. coli K12 and R2–R4 Strains. Materials 2021, 14, 2956. doi: 10.3390/ma14112956.

42.Paweł Kowalczyk, Monika Wilk, Parul Parul, Mateusz Szymczak, Karol Kramkowski, Stanisława Raj ,Grzegorz Skiba, Dorota Sulejczak, Patrycja Kleczkowska and Ryszard Ostaszewski. The Synthesis and Evaluation of Aminocoumarin Peptidomimetics as Cytotoxic Agents on Model Bacterial E. coli Strains. Materials 2021, 14, 5725. doi: 10.3390/ma14195725.

  1. Line 315-339, The protocols for MIC, MBC and Fpg digestion experiments are not clear, how to calculate the percent of digested DNA damage by Fpg enzyme.

Fpg protein digestion was performed according to the procedure prepared by the manufacturer New England Biolabs. The MIC and MBC tests used have been thoroughly described in our earlier papers which we cite in the literature section. We did not cite them in the full description to avoid self-plagiarism. The analyzed changes of topological forms of plasmid DNA after protein digestion were analyzed using the following programs: Imane Quant 5.2 and Microcal Origin. The percentage of cleavage of the known amount of DNA was estimated relative to the total and converted to percentage of damage relative to the uncut control. Approximately 4% of the bases in bacterial DNA have been found to be oxidized and recognized by the Fpg protein, which has been included in the relevant literature citations [37-45]. 

  1. Line 345-346, ‘E. coli K12 (without LPS in its structure)’, please confirm this information.

The K12 strain contains native LPS in its structure but its biological effect is negligible, therefore we have obtained the statement that it does not contain LPS in its structure. we backed this up with the appropriate manuscript quotation. Maciejewska, A.; Kaszowska, M.; Jachymek, W.; Lugowski, C.; Lukasiewicz, J. Lipopolysaccharide‐linked Enterobacterial Common Antigen (ECALPS) Occurs in Rough Strains of Escherichia coli R1, R2, and R4. Int. J. Mol. Sci. 2020, 21, 6038. Doi: 10.3390/ijms21176038.

  1. Line 427-428, Strain R4 was the most sensitive of all strains? How to obtain this result. According to Figure 3 and 4, these compounds showed the weakest activity against strain R4. Additionally, author stated that the longest lipopolysaccharide chain length of strain R4 makes it the most sensitive of all strains, please give more explanations for this.

We look at the degree of cytotoxicity of a given substance in relation to the analyzed MIC tests in given strains and in our opinion, the R4 strain which has the longest LPS is the most reactive because the length of the LPS determines its availability for the action of a given substance and modification of the bacterial membrane in which it is located, which induces oxidative stress in the cell visible precisely through changes in the LPS structure

  1. Line 446-451, The results showed that all analyzed compounds with different alkyl chain length and substituents containing a hydroxyl group or an amino group together with substituents containing bromine and chlorine can strongly change the topology of the plasmid form, even after digestion with Fpg protein. But why just fluorine group but not bromine and chlorine group were introduced into aromatic ring to evaluate the activity of the tested compounds. The authors can also further explore the role of bromine and chlorine group in the cellular reaction.

Fluorine is both small (van der Waals radius 1.47A ° ) and the most highly electronegative element in the Periodic Table (3.98 Pauling scale). This makes fluorine a unique atom that in turn has a profound effect when bound to carbon in small organic molecules. An examination of the role of fluorine in medicinal chemistry reveals that substitution of an organic compound with even a single fluorine atom or trifluoromethyl group located in a key position of a biologically active molecule can result in a profound pharmacological effect (Shah & Westwell, The role of fluorine in medicinal chemistry, Journal of Enzyme Inhibition and Medicinal Chemistry, October 2007; 22(5): 527–540DOI: 10.1080/14756360701425014). Currently, many fluorinated compounds are synthesized routinely in pharmaceutical research and are widely used in the treatment of diseases. Recently, we have shown that δ-lactones with a specific structure exhibit antimicrobial activity, the highest activity was observed for compounds containing  fluorine in their structure. Consequently, we decided to check whether the same effect would be visible for 3,4-dihydropyrimidin-2 (1H) -ones (DHPMs) modified in such a way that they contain a fluorine atom in their structure. Therefore, it was necessary to develop a convenient synthesis method

  1. Line 489-493, The cytotoxicity of DHPMs to bacterial DNA results from the modification of the components of the bacterial membrane and the LPS contained in it? It is hard to understand this sentence. Whether these compounds act by disrupting bacterial membrane? Please give experimental evidence for this. In addition, the relationship between DNA damage and membrane disruption is unclear.

Yes, they loosen the components of the cell membrane and tear it, getting into the genetic material of the cell, which leads to the formation of a specific oxidative stress, and therefore these compounds are tested for their suitability as potential antibiotics with better absorption and lower toxicity to the human body, but greater virulence for bacterial cells.

  1. Supplementary Materials for this manuscript are missing, please upload it.

 have been attached

  1. There are some grammatical errors in the context and figures. For example,

Line 40: remove ‘one’ corrected

Line 438: the position of ‘also’ seems insuitable  corrected

Line 442: please correct the form of ‘E.coli’ corrected

Line 456: ‘similarly’ should be substituted for ‘similar’ corrected

Line 518: please substitute ‘test’ for ‘tested’ corrected

Figure 8: please remove the ‘i’ after ‘enzyme’ and ‘antibiotics’

has been corrected in the drawing

Round 2

Reviewer 2 Report

My comments have been addressed.